# Addressing Uncertainty in LLMs to Enhance Reliability in Generative AI

**Ramneet Kaur[1], Colin Samplawski[1], Adam D. Cobb[1], Anirban Roy[1], Brian Matejek[1],**
**Manoj Acharya[1], Daniel Elenius[1], Alexander M. Berenbeim[2], John A. Pavlik[2],**
**Nathaniel D. Bastian[2], Susmit Jha[1]**
[1]Neuro-symbolic Computing and Intelligence, SRI, Menlo Park, USA
[2]Army Cyber Institute, United States Military Academy, West Point, NY USA
**Correspondence:** ramneet.kaur@sri.com

## Abstract

In this paper, we present a dynamic semantic clustering approach inspired by the Chinese Restaurant Process, aimed at addressing uncertainty in the inference of Large Language Models (LLMs). We quantify uncertainty of an LLM on a given query by calculating entropy of the generated semantic clusters. Further, we propose leveraging the (negative) likelihood of these clusters as the (non)conformity score within Conformal Prediction framework, allowing the model to predict a set of responses instead of a single output, thereby accounting for uncertainty in its predictions. We demonstrate the effectiveness of our uncertainty quantification (UQ) technique on two well-known question-answering benchmarks, COQA and TriviaQA, utilizing two LLMs—Llama-2-13b and Mistral-7b. Our approach achieves state-of-the-art (SOTA) performance in UQ, as assessed by metrics such as AUROC, AUARC, and AURAC. The proposed conformal predictor is also shown to produce smaller prediction sets while maintaining the same probabilistic guarantee of including the correct response, in comparison to existing SOTA conformal prediction baseline. Our code is publicly accessible at `https://shorturl.at/7yHSq`.

## 1 Introduction

Large language models (LLMs) are increasingly being utilized for various tasks, including open-world question answering. However, these models are known to exhibit hallucinations, confidently producing incorrect information or relying on faulty reasoning. Quantifying the uncertainty associated with an LLM for a given input offers a practical method to account for model's reliability on its response. When there is a strong correlation between LLM's accuracy and the computed uncertainty, one can effectively use uncertainty quantification (UQ) approach to determine when to place their trust in the model's responses.

The challenge of UQ for LLMs in generative context differs significantly from that encountered in regression or classification tasks. While the latter has been well-studied Guo et al. (2017); Jha et al. (2019); Xiao and Wang (2019); Hu and Khan (2021); Magesh et al. (2023), UQ for generative models presents unique challenges due to the free-form nature of the responses of varying lengths. Additionally, the syntactic similarity of generated sequences may not necessarily align with their semantic similarity.

Self-consistency theory can be employed to measure LLM's uncertainty on an input query by comparing semantic context of multiple outputs sampled from the LLM for that query (Wang et al., 2022).[1] Building on recent observations by Lin et al. (2023); Kuhn et al. (2023), we use embedding-

---

[1]Higher semantic consistency among multiple outputs would imply less uncertainty.

based semantic similarity to group sampled outputs into semantic clusters, and then quantify LLM's uncertainty by entropy of these clusters. Specifically, we propose a new dynamic semantic clustering algorithm based on the sequential distance dependent Chinese Restaurant Process (DDCRP) Blei and Frazier (2011); Tuncer and Schulz (2016) for quantifying LLM's uncertainty via entropy of the generated clusters.

Furthermore, we present a novel approach towards trustworthy inference from LLMs that combines the model's uncertainty with conformal prediction. Specifically, we propose to use (negative) likelihood of the clusters generated via DDCRP, as the (non)conformity score in the conformal prediction framework (Balasubramanian et al., 2014). This framework takes into account prediction uncertainty by generating a set instead of a single prediction, where the output set is guaranteed to contain the correct label (or answer in our case) with a certain confidence level. LLMs can generate free-form responses for question answering and so, the conformal prediction set in our approach is different from the usual setting of being a subset of known finite set of labels. We use semantic similarity to identify different or semantically diverse responses while constructing the conformal set. This use of conformal prediction by LLMs in question answering is the key novelty of our approach that aims to combine LLMs uncertainty on an input query with conformal prediction.

We demonstrate the efficacy of our UQ approach on two question-answering benchmarks, COQA and TriviaQA, using two LLMs, Llama-2-13b and Mistral-7b, achieving state-of-the-art (SOTA) results in most test cases using AUROC, AUARC and AURAC as metrics. We also show that our conformal predictor generates smaller prediction sets for the same probabilistic guarantees of including correct response compared to the SOTA conformal prediction baseline by Quach et al. (2023). This highlights not only the validity but also usefulness of our method in practical applications.

## 2    Related Work

**Uncertainty quantification (UQ)** for LLMs has received significant attention over the last few years. One approach is to explicitly query the model for the correctness probability (Kadavath et al., 2022). Another approach relies on utilizing the log-likelihood (Jiang et al., 2020) associated with its generated response by taking a product or an average or other statistical aggregation over the generated tokens. LLMs are known to be not well-calibrated (Mielke et al., 2022) and consequently, methods for calibrating LLMs (Huang et al., 2024) have also been proposed. Semantic entropy or predictive uncertainty that measures the (in)consistency among multiple responses has been proposed as a metric for UQ of LLMs (Kuhn et al., 2023; Lin et al., 2023). We also use semantic entropy for UQ but adopt a novel semantic clustering approach and empirically demonstrate its effectiveness.

**Conformal prediction (CP)** (Balasubramanian et al., 2014) has been used for deploying deep learning models (Kaur et al., 2022; Haroush et al., 2021; Kaur et al., 2023, 2024; Yang et al., 2024) in high-assurance applications wherein the model predicts a set instead of a single prediction such that one of the responses in the set is guaranteed to be correct with a probability higher than the user specified significance level. In the context of LLMs, conformal prediction has been used for providing coverage guarantees (Ye et al., 2024; Quach et al., 2023). Ye et al. (2024) concentrate on classification settings and propose non-conformity scores in the CP framework accordingly. In contrast, we focus on generative setting for LLMs in applications such as question-answering. Quach et al. (2023) propose generating diverse prediction sets based on the quality of individual responses, and a set scoring function. They utilize CP to derive hyperparameters ($\lambda$s) for diversity, quality, and set scoring function in their algorithm for coverage guarantees. We, instead propose, using (negative) likelihood of clusters representing semantically diverse responses, as the (non)conformity score in the CP framework for generating sets with coverage guarantees, and compare our results with Quach et al. (2023)'s approach.

## 3    Clustering by Semantic Equivalence

In this section, we introduce the proposed clustering approach, and then describe its usage in uncertainty quantification and building the conformal predictor for LLMs.

*Semantic entropy* over $|C|$ equivalence classes (or clusters) of multiple responses sampled from an LLM on an input query $x$ has been used as the *UQ measure* for the LLM on $x$ (Kuhn et al., 2023):

$$SE(x) = -\sum_c p(c|x) \log p(c|x),$$

where the probability of an equivalence class (corresponding to a cluster of embeddings), $c$, conditioned on $x$, is given by $p(c|x) = \sum_{\mathbf{s} \in c} p(\mathbf{s}|x)$ and $\mathbf{s} \in c$ denotes a sentence (or response) in an equivalence class $c \in C$. The probability of each sentence is given by the standard product of the conditional token probabilities:

$$p(\mathbf{s}|x) = \prod_i p(s_i|s_{<i}, x).$$

The result of using semantic entropy is to sharpen $p(c|x)$ when there are many responses with the same meaning, and therefore reduce the predicted entropy.

To generate clusters containing semantically equivalent responses, we need to define a function that checks semantic similarity between responses. Kuhn et al. (2023) choose the conservative approach of using Deberta (Natural Language Inference) model He et al. (2020) to only define two sentences to be semantically equivalent if and only if entailment was classified in both directions. Entailment in both directions is not necessary when one response includes additional information than another but they both contain the same relevant semantic information. For instance, let us consider the following input query: "What is the capital of France?" with the first response $\mathbf{s_1}$ = "Paris", and the second response $\mathbf{s_2}$ = "Paris, the capital city of France, is one of the most iconic and romantic cities in the world. It has a rich history dating back more than 2,000 years". Although, both the responses are semantically equivalent with respect to the input query, $\mathbf{s_2}$ entails $\mathbf{s_1}$ but vice versa is not true. Enforcing entailment in both directions can, therefore, lead to formation of more clusters than required with semantically equivalent information distributed in different clusters.

Thus, our approach computes the probability that two responses, $\mathbf{s}_i$ and $\mathbf{s}_j$, are in the same equivalence class (that is, the same semantic cluster), by taking the maximum entailment score output by Deberta, $p_{ij} = \max(p(\mathbf{s}_i \vdash \mathbf{s}_j), (p(\mathbf{s}_j \vdash \mathbf{s}_i))$. This choice of taking a maximum can be viewed as a quantitative disjunction of entailment in either direction. We then use this approach to build the score that a response, $\mathbf{s}_j$, belongs to an equivalence class, $c$, by using the average probability across all the cluster members:

$$w(\mathbf{s}_j \in c) = \frac{1}{|c|} \sum_{\mathbf{s}_i \in c} p_{ij} \tag{1}$$

This is also novel from the existing approaches for semantic clustering where $\mathbf{s_j}$ is assigned to the cluster $c$ if it is entailed in both directions by only one member of $c$. As observed in our qualitative evaluation (and reported in Appendix A.1), we postulate that this could be the reason for assigning semantically irrelevant responses to the same cluster because it is easier to incorrectly assign responses to the same cluster if we rely on only one member instead of all members in the cluster. In contrast, we take an average over all existing members of a cluster making our assignment more robust.

A naïve approach would be to greedily assign $\mathbf{s_j}$ to a cluster with the highest probability. However, this is not sufficient since we need a mechanism for forming new equivalence classes (or clusters). Given a set of responses, we need to decide when to form a new cluster and when to assign $\mathbf{s_j}$ to an existing cluster $c$ that has the highest score $w(\mathbf{s_j} \in c)$. We use the same mechanism as the Distance Dependent Chinese Restaurant Process (DDCRP) for iterative clustering (Blei and Frazier, 2011; Tuncer and Schulz, 2016). The motivation for using DDCRP is due to its probabilistic and dynamic nature for deciding on the formation of a new cluster or membership in an existing cluster based on semantic equivalence. Following DDCRP, the score for a sentence $\mathbf{s_j}$ in a new cluster $c^*$ is defined as:

$$w(\mathbf{s_j} \in c^*) = \frac{\alpha}{\alpha + |C|}, \tag{2}$$

where $|C|$ is the current number of clusters and $\alpha > 0$ is the rate parameter, i.e. prior over forming new clusters. The final (softmax) probabilities of assigning $\mathbf{s_j}$ to an existing cluster ($c_i$) and new

cluster ($c^*$) are respectively given by:

$$\text{Softmax}(z_i) = \frac{e^{z_i}}{\sum_j e^{z_j} + e^{z^*}}$$

$$\text{Softmax}(z^*) = \frac{e^{z^*}}{\sum_j e^{z_j} + e^{z^*}}, \tag{3}$$

where $z_i = w(s \in c_i)$, and $z^* = w(s \in c^*)$. Since the equivalence class assignment of the new response is related to semantic equivalence with the existing responses, we can map our clustering algorithm as an instance of sequential DDCRP.

Alg. 1 is the proposed clustering approach for iterative clustering of semantically equivalent responses. It is executed in a sequential fashion, starting with an empty set of clusters. After the first response from the LLM, we compute the score for forming a new cluster as: $\frac{\alpha}{\alpha+0} = 1$. This results in a new cluster probability of 1, leading a new cluster to be formed deterministically. In subsequent rounds, we compute the per cluster assignment scores for the new response as per Equations (1) and (2) for existing and new cluster respectively. We, then, compute the softmax probabilities for these scores from Equation (3), and assign the new response to a cluster (either existing cluster $c_i$ or a new cluster $c^*$) with the maximum probability.

---

**Algorithm 1** Clustering by Semantic Equivalence

1: **Input:** query $x$, LLM model $M$
2: **Parameter:** rate parameter $\alpha > 0$, number of responses $N$
3: **Output:** clusters $C$
4: **Initialize:** $C \leftarrow \emptyset$
5: **for** $i = 1$ to $N$ **do**
6:     $\mathbf{s}_i = M(x)$ {generate with LLM}
7:     $scores \leftarrow \mathbf{0}_{|C|+1}$
8:     **for** $c_j$ in $C$ **do**
9:         $scores[j] \leftarrow w(\mathbf{s}_i \in c_j)$
10:    **end for**
11:    $scores[-1] \leftarrow \frac{\alpha}{\alpha+|C|}$
12:    $probs \leftarrow \text{Softmax}(scores)$
13:    $k \leftarrow argmax(probs)$ {cluster assignment}
14:    **if** $k == |C| + 1$ **then**
15:        $C \leftarrow C \cup \{\mathbf{s}_i\}$ {new cluster}
16:    **else**
17:        $c_k \leftarrow c_k \cup \{\mathbf{s}_i\}$
18:    **end if**
19: **end for**
20:
21: **return** $C$

---

In all our experiments, we use the rate parameter of $\alpha = 0.5$. We performed a grid search over $\alpha \in [0.2, 0.3, \ldots, 0.7]$ and observed very little variance in performance. We set the value of number of responses $N$ is set to 20, which is consistent with the existing work (Kuhn et al., 2023; Lin et al., 2023; Quach et al., 2023).

## 3.1 Uncertainty Quantification

Different clusters containing semantically diverse responses to an input query can be used to quantify uncertainty of the LLM on the query. Similar to Kuhn et al. (2023), we also use entropy of the generated clusters from Alg. 1 as a measure of UQ for LLMs on a given query. Both qualitative and quantitative results indicate that the proposed clustering approach yields better results than Kuhn's baseline.

## 3.2 Conformal Set Prediction

For each cluster $c$, we have $p(c|x)$. We, therefore, use the negative log probability, $\log[1/p(c|x)]$, of the individual clusters, as the non-conformity score in conformal prediction (CP) framework (Balasubramanian et al., 2014), for generating prediction sets. Non-conformity scores of calibration datapoints is used to build a reference empirical distribution to compare against when building the prediction set. Specifically, depending on the desired significance level, $\epsilon$, prediction set is generated by comparing scores for the test clusters with a threshold from the empirical distribution: non-conformity score of calibration set at $(1-\epsilon)^{th}$ quantile of the distribution.[2] Intuitively, clusters with low negative log probability (or high likelihood) are more likely to be included in prediction sets compared to clusters with high negative log probability (or low likelihood). If an LLM outputs many semantically equivalent responses, then we expect the cluster's $\log[1/\sum_{\mathbf{s} \in c} p(\mathbf{s}|x)]$ to decrease due to the summation over the sentence probabilities by sharpening the cluster probability.

The use of CP for constructing the prediction sets gives us coverage guarantees on the true answer in the set with the probability greater than or equal to $1-\epsilon$ (Vovk et al., 2005). Alg. 2 is the proposed CP algorithm for generating prediction sets with coverage guarantees.

For an input query $x$ (from test or calibration set), clusters are generated via Alg. 1. We use negative log probability (nlp = $\log[1/p(c|x)]$) of each generated cluster ($c$) for $x$ as the non-conformity score for the cluster. For the desired significance level $\epsilon \in (0,1)$, the prediction threshold $\tau$ is decided as the score at $(1-\epsilon)^{th}$ quantile of the empirical distribution of non-conformity scores for the calibration clusters. Assuming all responses are semantically equivalent in a cluster, a single response from the test cluster ($c$) is added to the prediction set if its non-conformity score (nlp($c$)) is below the prediction threshold $\tau$. In our experiments, prediction set is constructed from the first response in the qualified test cluster.

---

**Algorithm 2** Conformal Prediction Sets by LLM

---

1: **Input:** query $x$, LLM model $M$, prediction threshold $\tau$ from $(1-\epsilon)^{th}$ quantile of the empirical distribution of calibration set non-conformity scores
2: **Output:** Prediction Set $\mathcal{O}$ with predictions on $x$ by $M$ s.t. $Pr.$(correct answer $\in \mathcal{O}) \geq 1-\epsilon$
3: $C$ = set of clusters from Alg. 1 on $(x, M)$
4: $S$ = set of non-conformity scores for the generated clusters: $\{\forall c \in C : \text{nlp}(c)\}$
5: $\mathcal{O} = \{$a response from $C[i]$ s.t. $S[i] \leq \tau : i = 1, \dots, |C|\}$
6: return $\mathcal{O}$

---

# 4 Experimental Results

The experimental evaluation focuses on two research questions. **RQ1:** Does the novel semantic clustering approach inspired from CRP improve the UQ of LLMs? **RQ2:** How does Alg 2 perform compared to the CP baseline on LLMs for free form generative responses?

**Datasets and Models**. We use two question-answer datasets: COQA (Reddy et al., 2019) and TriviaQA (Joshi et al., 2017), over which we compare the performance of two LLMs, Llama-2-13b: non-instruct model (Touvron et al., 2023), and Mistral-7b: instruct model (Jiang et al., 2023). Following existing literature (Lin et al., 2023; Kuhn et al., 2023), we deploy three evaluation methods: (1) We query GPT-4 (Achiam et al., 2023) by asking it to provide a rating on whether a response is correct with a value between 0 and 1, and label the response as correct if its rating $> 0.7$; (2) RougeL score (Lin, 2004) with a threshold $> 0.3$; (3) Deberta (He et al., 2020) to check for entailment of correct answer in the generated response.

## 4.1 UQ Performance

We report Area Under Accuracy-Rejection Curve (AUARC) (Nadeem et al., 2009), and Area Under Rejection-Accuracy Curve (AURAC) for comparing our performance on UQ with Kuhn et al. (2023)'s,

---

[2]Calibration set consists of only those clusters whose all responses are accurate.

| Model | Eval. | COQA Dataset | | | | TriviaQA Dataset | | | |
|---|---|---|---|---|---|---|---|---|---|
| | | Model Acc. | Sem. Ent. Unnorm/Norm | EigV | Ours Unnorm/Norm | Model Acc. | Sem. Ent. Unnorm/Norm | EigV | Ours Unnorm/Norm |
| Llama-13b | GPT-4 | 73.22 | 85.81/86.44 | **88.03** | 86.35/_87.47_ | 67.03 | 88.13/87.94 | **88.84** | 88.33/_88.54_ |
| Mistral-7b | GPT-4 | 73.38 | 81.91/82.68 | _82.82_ | 82.22/**82.95** | 60.68 | 80.99/_81.40_ | **82.03** | 81.23/**82.03** |
| Mean | GPT-4 | 73.30 | 83.86/84.56 | **85.43** | 84.29/_85.21_ | 63.86 | 84.56/84.67 | **85.44** | 84.78/_85.29_ |
| Llama-13b | RougeL | 72.75 | 86.03/87.05 | _87.92_ | 86.84/**88.34** | 64.60 | 85.62/85.19 | 85.76 | _85.86_/**85.87** |
| Mistral-7b | RougeL | 44.74 | _64.37_/62.93 | 63.43 | **64.60**/63.48 | 42.33 | _70.18_/68.13 | 69.41 | **70.26**/68.81 |
| Mean | RougeL | 58.75 | 75.20/74.99 | 75.65 | _75.72_/**75.91** | 53.47 | _77.90_/76.66 | 77.59 | **78.06**/77.34 |
| Llama-13b | Deberta | 63.74 | 80.21/79.48 | **82.68** | 81.04/_81.37_ | 63.33 | 84.92/84.34 | **85.60** | _85.23_/85.13 |
| Mistral-7b | Deberta | 11.23 | _23.56_/20.71 | 20.88 | **23.53**/21.05 | 33.92 | _62.29_/59.53 | 60.39 | **62.37**/60.16 |
| Mean | Deberta | 37.49 | _51.89_/50.10 | 51.78 | **52.29**/51.21 | 48.63 | _73.61_/71.94 | 73.00 | **73.80**/72.65 |

Table 1: AUARC (↑) results in comparison to Kuhn et al. (2023)'s Semantic Entropy (Sem. Ent.) UQ metric, and SOTA EigV metric by Lin et al. (2023). Best results are in bold and second best are underlined.

| Model | Eval. | COQA Dataset | | | | TriviaQA Dataset | | | |
|---|---|---|---|---|---|---|---|---|---|
| | | Model Acc. | Sem. Ent. Unnorm/Norm | EigV | Ours Unnorm/Norm | Model Acc. | Sem. Ent. Unnorm/Norm | EigV | Ours Unnorm/Norm |
| Llama-13b | GPT-4 | 73.22 | 58.97/56.90 | **54.63** | 58.42/_55.32_ | 67.03 | 40.09/40.27 | _39.42_ | 39.92/**39.38** |
| Mistral-7b | GPT-4 | 73.38 | 63.06/62.02 | **59.83** | 62.77/_61.41_ | 60.68 | 35.57/35.04 | **33.19** | 35.13/_33.29_ |
| Mean | GPT-4 | 73.30 | 61.02/59.46 | **57.23** | 60.60/_58.37_ | 63.86 | 37.83/37.66 | **36.31** | 37.53/_36.34_ |
| Llama-13b | RougeL | 72.75 | 56.75/55.53 | _53.78_ | 55.78/**52.65** | 64.60 | 39.12/39.39 | 38.93 | _38.81_/**38.35** |
| Mistral-7b | RougeL | 44.74 | 27.62/29.65 | **27.12** | 27.37/28.26 | 42.33 | 17.15/19.56 | _17.06_ | **16.95**/18.11 |
| Mean | RougeL | 58.75 | 42.19/42.59 | **40.45** | 41.58/_40.46_ | 53.47 | 28.14/29.48 | _28.00_ | **27.88**/28.23 |
| Llama-13b | Deberta | 63.74 | 46.07/46.91 | **42.04** | 45.07/_43.56_ | 63.33 | 37.23/37.94 | **36.70** | 36.88/36.84 |
| Mistral-7b | Deberta | 11.23 | _3.84_/5.70 | 4.13 | **3.82**/5.00 | 33.92 | _11.00_/13.54 | 11.35 | **10.89**/12.45 |
| Mean | Deberta | 37.49 | 24.96/26.31 | **23.09** | _24.45_/24.28 | 48.63 | 24.12/25.74 | _24.03_ | **23.89**/24.65 |

Table 2: AURAC (↓) results in comparison to Kuhn et al. (2023)'s Semantic Entropy (Sem. Ent.) UQ metric, and SOTA EigV metric by Lin et al. (2023). Best results are in bold and second best are underlined.

and Lin et al. (2023)'s approaches. For Lin et al. (2023)'s approach, we use EigV as their UQ metric[3]. While AUARC has been used as an evaluation metric previously (Lin et al., 2023), we include AURAC as a new metric. AUARC is an indicator of the accuracy of accepted (or highly certain) samples, and AURAC is an indicator of the accuracy of rejected (or highly uncertain) samples. In addition to AUARC, AURAC also indicates calibration of the UQ metric: we would like the accuracy of the model on the rejected samples by the UQ metric to be as low as possible, i.e. not rejecting samples on which LLMs are accurate.

The results are reported in Tables 1, and 2. Similar to Kuhn et al. (2023)'s, we also report our results with the UQ score unnormalized (Unnorm)/normalized (Norm) on the response's length. We outperform the baseline by Kuhn et al. (2023) in all the test cases, indicating that the proposed clustering approach performs better in UQ. We achieve competitive results in comparison to the current SOTA by Lin et al. (2023) by outperforming them in most cases. We also report AUROC, and compare with other baselines in Appendix A.2.

## 4.2 Conformal Prediction Results

The desired properties of a prediction set is that the accuracy of the set should be as high as possible with a smaller set size. So, here we report accuracy and set size as the evaluation metrics.

### 4.2.1 Comparison with the CP Baseline

Fig. 1 shows Alg. 2 results in comparison with the existing baseline by Quach et al. (2023) on using CP for generating prediction sets with coverage guarantees.

---

[3]They also propose 'Ecc', and 'Deg' as other UQ metrics. Consistent with their paper, we found that the best results in most of the cases are with 'EigV' metric, and therefore we compare our results with this UQ metric.

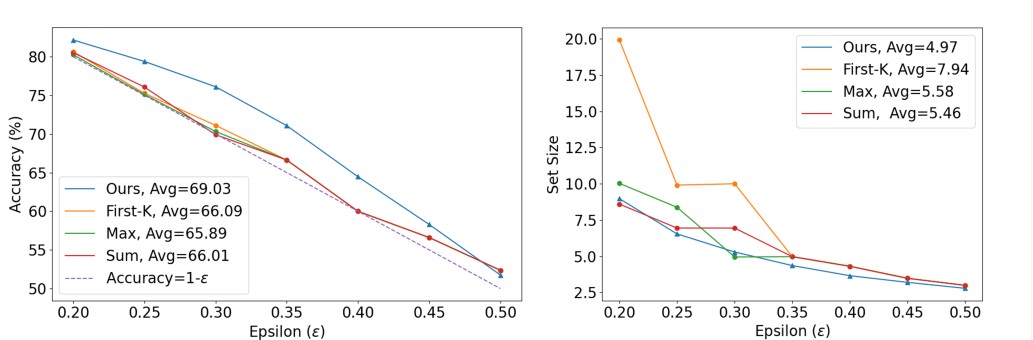

Figure 1: Comparison of Accuracy (left) and Set Size of prediction sets (right) with Conformal Prediction baseline (Quach et al., 2023).

The coverage guarantee (or guarantee of the correct answer contained in the prediction set by CP) is expected to be $\geq (1 - \epsilon)$. So, as the value of $\epsilon$ increases, the accuracy and the set size is expected to decrease. This is what we observe for both approaches: ours and the baseline, with both the approaches satisfying the coverage guarantees. Quach et al. (2023) report results with different variations of their proposed algorithm (Algorithm 1 of their paper) in terms of the set scoring function ($\mathcal{F}$): First-$K$, Max, and Sum, and on **TriviaQA with Llama-2-13b**. We outperform these results for all the three variations on both evaluation metrics.

### 4.2.2 Experiments on COQA

We also evaluate our Alg. 2's performance on COQA. Figure 2 shows these results for both accuracy and set size and with all the three GT evaluation approaches on COQA: GPT-4, RougeL, and Deberta.

Here, we also report the point accuracy, which is the average accuracy of the individual $N = 20$ generations. For $\epsilon \leq 0.35$, the prediction set accuracy is always higher than the point accuracy. Consistent with the results on TriviQA, here also the value of accuracy and set size decreases with the increase in the value of $\epsilon$. GPT and RougeL evaluations satisfy the coverage guarantee $\forall \epsilon \geq 0.15$. Even though conformal prediction provides a rigorous theoretical guarantee, deviations from the coverage guarantee can occur in practice due to limited sample variability in the caliibration set (Angelopoulos and Bates, 2021). This justifies the accuracy results with Deberta Evaluation and the other two evaluations with $\epsilon < 0.15$.

One difference to note here from the TriviaQA experiments is in the set of $\epsilon$ values. For TriviaQA, we reported results with $\epsilon \in \{0.2, \ldots, 0.5\}$, and for COQA we reported results with $\epsilon \in \{0.1, 0.2, \ldots, 0.5\}$. This is because we were getting 'nan' at $\epsilon = 0.1$ from the Quach's baseline on TriviaQA. While further investigation on their code, we figured it out that they are hard-coding the value as 'nan' where the search for their algorithm's hyperparameters ($\lambda$s) might be failing.

## 5 Conclusion

This paper takes a step towards enhancing reliability in generative AI by addressing uncertainty in LLMs on a given query. Our approach focuses on the semantic equivalence of responses generated by an LLM when prompted multiple times for the input query. It is based on the idea that an LLM is expected to be accurate if it consistently generates semantically similar outputs when prompted multiple times with the same input. The underlying assumption here is that consistency can serve as an indicator of accuracy. Testing this hypothesis is one of the future works that we intend to investigate. Additionally, we aim to explore alternative scoring methods beyond the probability of response—calculated as the product of conditional token probabilities— used to determine the likelihood of semantic clusters. This is important because response probability can be sensitive to its length, which poses a significant challenge when dealing with free-form generations produced by LLMs.

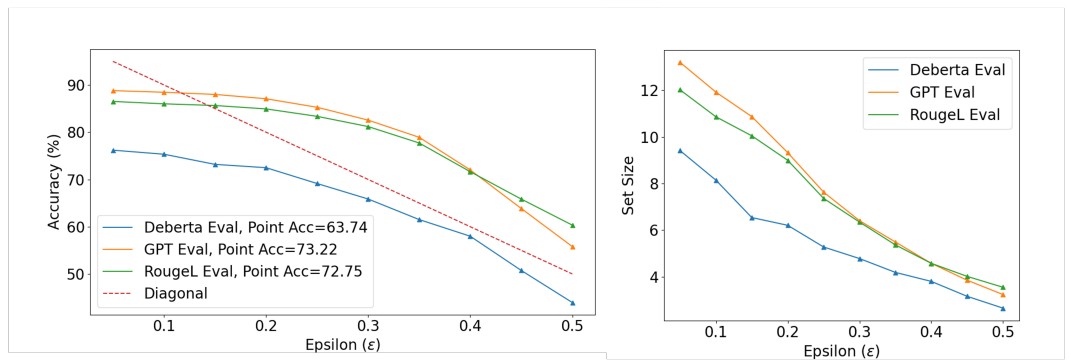

Figure 2: Alg. 2's Accuracy (left) and Set Size (right) evaluation on COQA for Llama-13b.

## Acknowledgments

This work was supported in part by the United States Air Force and Defense Advanced Research Projects Agency (DARPA) under Contract No.FA8750-23-C-0519, the Defense Advanced Research Projects Agency (DARPA) under Agreement No. HR0011-24-9-0424 and Defense Logistics Agency (DLA) and the Advanced Research Projects Agency for Health (ARPA-H) under Contract Number SP4701-23-C-0073. Any opinions, findings and conclusions or recommendations expressed in this material are those of the authors and do not necessarily reflect the Department of Defense, DARPA, DLA, ARPA-H or the United States Government.

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

# A  Appendix

## A.1  Qualitative comparison of Clustering Approaches

Here, we provide an example from both COQA and TriviaQA datasets to analyse how our new clustering approach compares to the original approach by Kuhn et al. (2023). We look at the quality of clusters formed by both approaches.

### A.1.1  An example from COQA Dataset

**Story**: CHAPTER XXXIV Arthur remained at the gate while Ruth climbed Maria's front steps. She heard the rapid click of the type-writer, and when Martin let her in, found him on the last page of a manuscript. She had come to make certain whether or not he would be at their table for Thanksgiving dinner; but before she could broach the subject Martin plunged into the one with which he was full. "Here, let me read you this," he cried, separating the carbon copies and running the pages of manuscript into shape. "It's my latest, and different from anything I've done. It is so altogether different that I am almost afraid of it, and yet I've a sneaking idea it is good. You be judge. It's an Hawaiian story. I've called it 'Wiki-wiki'." His face was bright with the creative glow, though she shivered in the cold room and had been struck by the coldness of his hands at greeting. She listened closely while he read, and though he from time to time had seen only disapprobation in her face, at the close he asked:- "Frankly, what do you think of it?" "I–I don't know," she, answered. "Will it–do you think it will sell?" "I'm afraid not," was the confession. "It's too strong for the magazines. But it's true, on my word it's true." "But why do you persist in writing such things when you know they won't sell?" she went on inexorably. "The reason for your writing is to make a living, isn't it?"
**Question**: 'Did he answer her?'
**Answer**: 'No'
**Generated Responses from Llama-13b**: ['Yes', 'Yes', "He didn't", 'He did, only not directly', 'No', 'No', 'No', 'He asked her what she thought', 'He told her his latest story', 'Yes', 'No', 'A sneaking yes', 'He ran the manuscript up to Miss Lawton', 'No', 'In the affirmative', 'Yes', 'No', 'No', 'Yes', 'Yes']

**Results**: Figures 3, and 4 show the clusters formed by Kuhn et al. (2023), and our approach respectively. For brevity, we include only unique responses in a cluster. As it can be seen, Kuhn et al. (2023) approach puts semantically different responses in the same cluster (responses 3, 4, and 6 in cluster 1 for 'Yes'), whereas ours separate them out in different clusters.

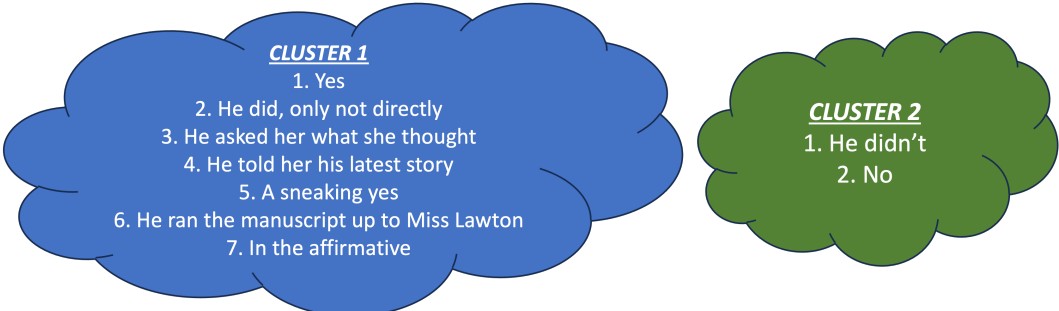

Figure 3: Clusters generated by  Kuhn et al. (2023)'s approach on COQA example.

### A.1.2  An example from TriviaQA Dataset

**Question**: What is 'The Old Lady of Threadneedle Street'?
**Answer**: Bank of England
**Generated Responses from Llama-13b**: ['Bank of England', 'Bank of England', 'Bank of England', 'The Bank of England', 'A nickname; what was it really?', 'Bank of England', 'The Bank of England', 'The Bank of England', 'The Bank of England', 'The Bank of England', 'The Bank of England', 'Bank of England', 'The Bank of England', 'Bank of England', 'The Bank of England', 'Bank of England', 'Bank of England', 'Bank of England', 'The Bank of England', 'Bank Of England']

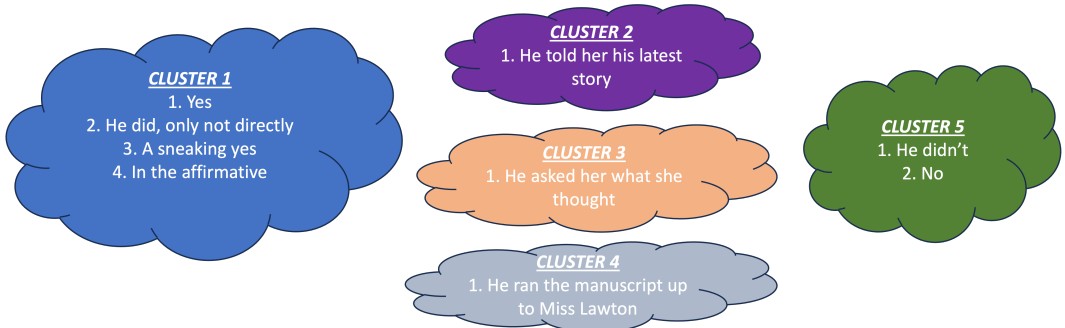

Figure 4: Clusters generated by our approach (Alg. 1) on COQA example.

**Results**: Figures 3, and 4 show the clusters formed by Kuhn et al. (2023), and our approach respectively. Again for brevity, we include only unique responses in a cluster. As it can be seen, Kuhn et al. (2023) approach puts semantically different responses in the same cluster (response 3 in cluster 1 for 'Bank of England'), whereas ours separate them out in different clusters.

We observed similar results on other stories from COQA, and questions from TriviaQA datasets.

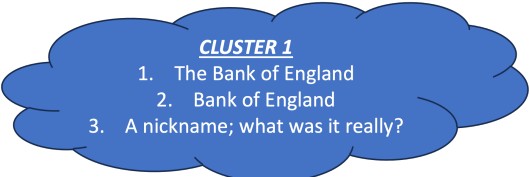

Figure 5: Clusters generated by Kuhn et al. (2023)'s approach on TriviaQA example.

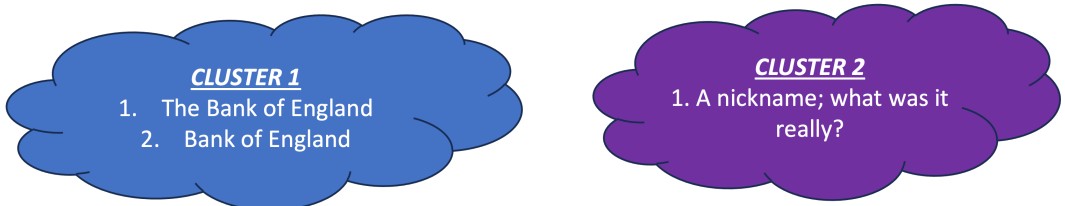

Figure 6: Clusters generated by our approach (Alg. 1) on TriviaQA example.

## A.2  All UQ results

Here, we include all results on UQ performance from Section 4.1: comparison with additional baselines on AUARC, AURAC and AUROC evaluation metrics. In addition to Kuhn et al. (2023)'s Sem. Ent. (Unnorm/Norm), and (Lin et al., 2023)'s EigV results reported in the main paper, we include "Numset", "LexiSim", and "SelfProb" baselines here. Numset uses the number of semantic sets (or clusters) as the UQ metric, and has been previously used in (Lin et al., 2023) as one of the baselines. Higher the numset, more uncertain is the LLM on the input query. LexiSim uses the average of RougeL distance between every pair of responses for UQ. Here, higher the Lexisim, lower is the uncertainty. Again, Lexisim has been used as a baseline by Kuhn et al. (2023), and Lin et al. (2023). SelfProb (Kadavath et al., 2022) estimates if the probability of a model's response is correct by asking the model itself, and use that as the UQ metric. We follow the same prompt format as Lin et al. (2023) for asking the model about the probability, and report average over all responses. Here, higher the SelfProb, lower is the uncertainty.

Tables 3, 5, and 7 are AUARC, AUROC, and AURAC are results on COQA. And Tables 4, 6, and 8 are AUARC, AUROC, and AURAC are results on TriviaQA. Again, we report all results from the three GT evaluation methods: GPT-4, RougeL, and Deberta. We achieve either the best or second best in all but two test cases.

| Model | GT | Model Acc | Sem. Ent. Unnorm/Norm | NumSet | LexiSim | SelfProb | EigV | Ours Unnorm/Norm |
|---|---|---|---|---|---|---|---|---|
| Llama-13b | GPT-4 | 73.22 | 85.81/86.44 | 79.78 | 86.14 | 75.50 | **88.03** | 86.35/87.47 |
| Mistral-7b | GPT-4 | 73.38 | 81.91/82.68 | 75.63 | 81.73 | **85.14** | 82.82 | 82.22/82.95 |
| Mean | GPT-4 | 73.30 | 83.86/84.56 | 77.71 | 83.94 | 80.32 | **85.43** | 84.29/85.21 |
| Llama-13b | RougeL | 72.75 | 86.03/87.05 | 77.79 | 88.17 | 73.35 | 87.92 | 86.84/**88.34** |
| Mistral-7b | RougeL | 44.74 | 64.37/62.93 | 46.99 | 59.61 | 52.64 | 63.43 | **64.60**/63.48 |
| Mean | RougeL | 58.75 | 75.20/74.99 | 62.39 | 73.89 | 63.00 | 75.65 | 75.72/**75.91** |
| Llama-13b | Deberta | 63.74 | 80.21/79.48 | 69.36 | 79.02 | 65.23 | **82.68** | 81.04/81.37 |
| Mistral-7b | Deberta | 11.23 | 23.56/20.71 | 11.63 | 16.70 | 12.21 | 20.88 | **23.53**/21.05 |
| Mean | Deberta | 37.49 | 51.89/50.10 | 40.50 | 47.86 | 38.72 | 51.78 | **52.29**/51.21 |

Table 3: AUARC (↑) results in comparison to all baselines on **COQA**. Best results are in bold and second best are underlined.

| Model | GT | Model Acc | Sem. Ent. Unnorm/Norm | NumSet | LexiSim | SelfProb | EigV | Ours Unnorm/Norm |
|---|---|---|---|---|---|---|---|---|
| Llama-13b | GPT-4 | 67.03 | 88.13/87.94 | 83.84 | 84.52 | 73.09 | **88.84** | 88.33/88.54 |
| Mistral-7b | GPT-4 | 60.68 | 80.99/81.40 | 74.72 | 76.65 | **84.46** | 82.03 | 81.23/82.03 |
| Mean | GPT-4 | 63.86 | 84.56/84.67 | 79.28 | 80.59 | 78.78 | **85.44** | 84.78/85.29 |
| Llama-13b | RougeL | 64.60 | 85.62/85.19 | 79.75 | 84.01 | 70.34 | 85.76 | 85.86/**85.87** |
| Mistral-7b | RougeL | 42.33 | 70.18/68.13 | 54.53 | 61.72 | 62.03 | 69.41 | **70.26**/68.81 |
| Mean | RougeL | 53.47 | 77.90/76.66 | 67.14 | 72.87 | 66.19 | 77.59 | **78.06**/77.34 |
| Llama-13b | Deberta | 63.33 | 84.92/84.34 | 79.11 | 80.01 | 68.04 | **85.60** | 85.23/85.13 |
| Mistral-7b | Deberta | 33.92 | 62.29/59.53 | 44.88 | 51.80 | 50.33 | 60.39 | **62.37**/60.16 |
| Mean | Deberta | 48.63 | 73.61/71.94 | 62.00 | 65.91 | 59.19 | 73.00 | **73.80**/72.65 |

Table 4: AUARC (↑) results in comparison to all baselines on **TriviaQA**. Best results are in bold and second best are underlined.

| Model | GT | Model Acc | Sem. Ent. Unnorm/Norm | NumSet | LexiSim | SelfProb | EigV | Ours Unnorm/Norm |
|---|---|---|---|---|---|---|---|---|
| Llama-13b | GPT-4 | 73.22 | 85.19/88.69 | 73.06 | 82.63 | 53.74 | **92.83** | 87.87/91.90 |
| Mistral-7b | GPT-4 | 73.38 | 79.91/81.99 | 58.00 | 76.57 | 79.46 | 82.77 | 81.48/**84.07** |
| Mean | GPT-4 | 73.3 | 82.55/85.34 | 65.53 | 79.6 | 66.6 | 87.80 | 84.68/**87.99** |
| Llama-13b | RougeL | 72.75 | 80.87/86.48 | 68.24 | **92.33** | 48.40 | 88.45 | 83.90/90.28 |
| Mistral-7b | RougeL | 44.74 | 83.52/83.97 | 55.43 | 83.15 | 70.62 | **86.84** | 84.63/85.69 |
| Mean | RougeL | 58.75 | 82.20/85.23 | 61.84 | 87.74 | 59.51 | 87.65 | 84.27/**87.99** |
| Llama-13b | Deberta | 63.74 | 85.59/88.80 | 69.55 | 87.45 | 48.92 | **93.09** | 88.38/92.02 |
| Mistral-7b | Deberta | 11.23 | 93.98/91.22 | 54.62 | 91.57 | 59.48 | 93.85 | **94.25**/92.07 |
| Mean | Deberta | 37.485 | 89.79/90.01 | 62.09 | 89.51 | 54.2 | **93.47** | 91.32/92.05 |

Table 5: AUROC (↑) results in comparison to all baselines on **COQA**. Best results are in bold and second best are underlined.

| Model | GT | Model Acc | Sem. Ent. Unnorm/Norm | NumSet | LexiSim | SelfProb | EigV | Ours Unnorm/Norm |
|---|---|---|---|---|---|---|---|---|
| Llama-13b | GPT-4 | 67.03 | 94.74/96.85 | 92.04 | 86.64 | 59.38 | **97.48** | 95.29/97.39 |
| Mistral-7b | GPT-4 | 60.68 | 90.58/93.59 | 80.93 | 81.00 | **94.75** | 93.66 | 91.48/94.24 |
| Mean | GPT-4 | 63.86 | 92.66/95.22 | 86.49 | 83.82 | 77.07 | 95.57 | 93.39/**95.82** |
| Llama-13b | RougeL | 64.60 | 92.63/94.50 | 88.59 | 97.01 | 59.84 | **95.25** | 93.27/95.23 |
| Mistral-7b | RougeL | 42.33 | 95.26/94.80 | 74.87 | 94.66 | 84.67 | **96.34** | 95.61/95.36 |
| Mean | RougeL | 53.47 | 93.95/94.65 | 81.73 | **95.84** | 72.26 | 95.80 | 94.44/95.30 |
| Llama-13b | Deberta | 63.33 | 92.61/94.34 | 87.73 | 86.58 | 55.96 | **96.55** | 93.49/95.30 |
| Mistral-7b | Deberta | 33.92 | 96.55/94.90 | 73.84 | 93.06 | 82.01 | 96.64 | **96.73**/95.30 |
| Mean | Deberta | 48.63 | 94.58/94.62 | 80.79 | 89.82 | 68.99 | **96.60** | 95.11/95.30 |

Table 6: AUROC (↑) results in comparison to all baselines on **TriviaQA**. Best results are in bold and second best are underlined.

| Model | GT | Model Acc | Sem. Ent. Unnorm/Norm | NumSet | LexiSim | SelfProb | EigV | Ours Unnorm/Norm |
|---|---|---|---|---|---|---|---|---|
| Llama-13b | GPT-4 | 73.22 | 58.97/56.90 | 63.45 | 61.35 | 72.29 | **54.63** | 58.42/55.32 |
| Mistral-7b | GPT-4 | 73.38 | 63.06/62.02 | 67.19 | 62.21 | 61.18 | **59.83** | 62.77/61.41 |
| Mean | GPT-4 | 73.30 | 61.02/59.46 | 65.32 | 61.78 | 66.74 | **57.23** | 60.60/58.37 |
| Llama-13b | RougeL | 72.75 | 56.75/55.53 | 65.30 | 58.51 | 73.12 | 53.78 | 55.78/**52.65** |
| Mistral-7b | RougeL | 44.74 | 27.62/29.65 | 40.13 | 34.11 | 33.56 | **27.12** | 27.37/28.26 |
| Mean | RougeL | 58.75 | 42.19/42.59 | 52.72 | 46.31 | 53.34 | **40.45** | 41.58/40.46 |
| Llama-13b | Deberta | 63.74 | 46.07/46.91 | 55.50 | 53.07 | 63.32 | **42.04** | 45.07/43.56 |
| Mistral-7b | Deberta | 11.23 | 3.84/5.70 | 9.84 | 11.83 | 9.45 | 4.13 | **3.82**/5.00 |
| Mean | Deberta | 37.49 | 24.96/26.31 | 32.67 | 32.45 | 36.39 | **23.09** | 24.45/24.28 |

Table 7: AURAC (↓) results in comparison to all baselines on **COQA**. Best results are in bold and second best are underlined.

| Model | GT | Model Acc | Sem. Ent. Unnorm/Norm | NumSet | LexiSim | SelfProb | EigV | Ours Unnorm/Norm |
|---|---|---|---|---|---|---|---|---|
| Llama-13b | GPT-4 | 67.03 | 40.09/40.27 | 43.03 | 54.28 | 60.31 | 39.42 | 39.92/**39.38** |
| Mistral-7b | GPT-4 | 60.68 | 35.57/35.04 | 40.02 | 47.45 | 33.71 | **33.19** | 35.13/33.29 |
| Mean | GPT-4 | 63.86 | 37.83/37.66 | 41.525 | 50.87 | 47.01 | **36.31** | 37.53/36.34 |
| Llama-13b | RougeL | 64.60 | 39.12/39.39 | 42.74 | 50.56 | 58.41 | 38.93 | 38.81/**38.35** |
| Mistral-7b | RougeL | 42.33 | 17.15/19.56 | 25.09 | 32.12 | 21.46 | 17.06 | **16.95**/18.11 |
| Mean | RougeL | 53.47 | 28.14/29.48 | 33.915 | 41.34 | 39.94 | 28.00 | **27.88**/28.23 |
| Llama-13b | Deberta | 63.33 | 37.23/37.94 | 40.09 | 53.01 | 58.23 | **36.70** | 36.88/36.84 |
| Mistral-7b | Deberta | 33.92 | 11.00/13.54 | 18.51 | 27.93 | 15.97 | 11.35 | **10.89**/12.45 |
| Mean | Deberta | 48.63 | 24.12/25.74 | 29.3 | 40.47 | 37.1 | 24.03 | **23.89**/24.65 |

Table 8: AURAC (↓) results in comparison to all baselines on **TriviaQA**. Best results are in bold and second best are underlined.

