# OpenReview forum: "Addressing Uncertainty in LLMs to Enhance Reliability in Generative AI"
_NeurIPS.cc/2024/Workshop/SafeGenAi — SafeGenAi Poster_

### Official Review · Reviewer_yB9E · 2024-10-08

**Rating:** 7
**Confidence:** 5

**Review:**

Summary: The paper presents a novel approach to uncertainty quantification (UQ) in large language models (LLMs), specifically for generative tasks like open-world question answering. The authors propose leveraging embedding-based semantic similarity to group outputs into clusters, using a dynamic semantic clustering algorithm based on the Chinese Restaurant Process (DDCRP). Additionally, they integrate conformal prediction with this clustering method to generate reliable prediction sets. Empirical results show that their approach produces smaller, more precise prediction sets while maintaining high accuracy.

Strengths:
1. The writing is clear and readable, supported by ample visualizations throughout the main text and appendix.
2. The main contribution is well-defined: utilizing semantic similarity to assess uncertainty. The authors clearly explain the motivation and rationale behind this approach.
3. The empirical results are compelling. Comparisons with different baselines and datasets effectively demonstrate the effectiveness of the dynamic semantic clustering and conformal prediction methods.

Weaknesses:
1. The approach lacks a theoretical guarantee such as marginal coverage, which is a standard in conformal prediction.
2. The paper focuses solely on the question-answering task, but this approach has the potential for broader applications such as natural language generation and text classification, which are not explored in details.

---

### Official Review · Reviewer_qbdC · 2024-10-09
**This paper proposes a dynamic semantic clustering approach inspired by the Chinese Restaurant Process to address uncertainty in the inference of LLMs. The authors quantify uncertainty with the entropy of semantic clusters and integrate it with a conformal prediction framework to produce prediction sets that account for model uncertainty. Their experimental results show that the approach achieves state-of-the-art performance in uncertainty quantification metrics and generates more compact prediction sets than existing conformal prediction baselines.**

**Rating:** 6
**Confidence:** 4

**Review:**

The main contributions of the paper include: 1) introducing a novel clustering method based on the sequential distance-dependent Chinese Restaurant Process for uncertainty quantification in LLMs; 2) leveraging negative likelihood within a conformal prediction framework to improve set prediction accuracy; and 3) validating the proposed method's effectiveness through experiments on question-answering benchmarks, demonstrating superior performance on UQ metrics like AUROC and AURAC compared to existing methods​.

For strengths: 1) The idea of using dynamic semantic clustering inspired by the Chinese Restaurant Process to quantify uncertainty is innovative. Most existing work focuses on static methods of uncertainty estimation or purely probabilistic models. However, these approaches cannot adaptively group semantically similar predictions or fully capture the complexities of model uncertainty, which could limit their effectiveness in generating accurate prediction sets. This paper introduces a method that utilizes entropy-based clustering combined with a conformal prediction framework, as well as enhances the precision and compactness of uncertainty quantification in LLMs. 2) The workflow is well-structured, as it integrates dynamic clustering with negative likelihood estimation to make the uncertainty quantification process more adaptive and responsive to semantic variations in predictions. This approach provides a more nuanced understanding of prediction uncertainty and further enhances the interpretability and reliability of LLM outputs in high-stakes applications. 3) The experiments are extensive, with detailed analysis of the results. These experiments validate the effectiveness of the dynamic clustering and conformal prediction framework and demonstrate the robustness and adaptability of the method across different datasets and varying levels of model complexity​.

For weaknesses: 1) In Section 3.1, Methodology, the authors introduce the adaptation of the Chinese Restaurant Process (CRP) for semantic clustering, but they do not provide sufficient detail on how the parameters of the CRP are tuned or optimized. Also there is no clear explanation of how distance-dependent clustering ensures convergence or how sensitive the method is to changes in cluster formation parameters. 2) The paper emphasizes its dynamic semantic clustering approach, but the authors do not compare comprehensively with traditional uncertainty estimation methods such as Bayesian neural networks or Monte Carlo Dropout (Section 5.2, Experimental Results). The paper could benefit from adding more detailed comparisons with existing methods and further discussion of prediction accuracy or computational efficiency